

# Performance comparison of QuantiFERON-TB Gold In-Tube and QuantiFERON-TB Gold Plus in detecting *Mycobacterium tuberculosis* infection among HIV patients in China

Peng Lu[1,2,*], Haitao Yang[1,*], Fang Ge[3], Kai Wu[3], Yilin Lian[4], Xiaoyan Ding[1], Jingjing Pan[1], Hui Ding[1], Wei Lu[1], Qiao Liu[1] and Limei Zhu[1]

[1] Department of Chronic Communicable Disease, Jiangsu Provincial Center for Disease Control and Prevention, Nanjing, China
[2] Department of Epidemiology, Key Laboratory of Public Health Safety and Emergency Prevention and Control Technology of Higher Education Institutions in Jiangsu Province, School of Public Health, Nanjing Medical University, Nanjing, China
[3] Central Hospital, Jiangsu Prison Administration, Changzhou, China
[4] School of Public Health, Southeast University, Nanjing, China
[*] These authors contributed equally to this work.

## ABSTRACT

**Introduction.** No direct comparative study assessing QuantiFERON-TB Gold In-Tube (QFT-GIT) and QuantiFERON-TB Gold Plus (QFT-Plus) for *Mycobacterium tuberculosis* infection among persons living with HIV (PLHIV) in China has been conducted.

**Methods.** Simultaneous QFT-GIT and QFT-Plus tests were conducted on PLHIV in a prison hospital. Positivity and negativity results from both assays were compared, and their diagnostic agreement was assessed.

**Results.** A total of 232 PLHIV individuals were included in this study. Among them, 57 patients (24.6%) and 56 patients (24.1%) were diagnosed with *Mycobacterium tuberculosis* infection based on QFT-GIT results and QFT-Plus, respectively. The overall agreement between the two assays was 98.3%, with a Cohen's kappa value of 0.954. Consistency rates were observed between QFT-GIT plus, QFT-Plus TB1 and TB2 with QFT-GIT were 98.3%, 97.4% and 97.8%. The IFN-γ levels measured in QFT-GIT were found to surpass those in QFT-Plus TB1 ($P = 0.04$), while the difference compared to QFT-Plus TB2 exhibited a marginal trend ($P = 0.134$). Among the subgroup of 52 individuals who underwent dual QFT-GIT tests, a significant proportion of 23.1% (12 individuals) experienced a change in their QFT-GIT results, transitioning from a positive to a negative outcome.

**Conclusions.** The diagnostic performance of QFT-GIT and QFT-Plus for *Mycobacterium tuberculosis* infection among PLHIV with relatively higher CD4 counts was found to be comparable. Additionally, our investigation revealed that irrespective of the treatment regimen, whether it involved chemotherapy or immunotherapy, preventive *Mycobacterium tuberculosis* infection interventions among PLHIV consistently led to a reduction in IFN-γ levels.

Corresponding authors
Qiao Liu, liuqiaonjmu@163.com
Limei Zhu, lilyam0921@163.com

## INTRODUCTION

Approximately one-quarter of the world's population is believed to have *Mycobacterium tuberculosis* infection, and within this group, 5% to 10% will develop active tuberculosis over the course of their lives (*World Health Organization, 2020*). In pursuit of the 2035 goal outlined in the End TB Strategy to reduce the tuberculosis incidence rate by 90%, the World Health Organization (WHO) suggests the screening and preventive treatment of *Mycobacterium tuberculosis* infection among populations at elevated risk including persons living with HIV (PLHIV) (*Uplekar et al., 2015*). The WHO recommends both the tuberculin skin test (TST) and the interferon gamma release assay (IGRA) for diagnosing *Mycobacterium tuberculosis* infection (*World Health Organization, 2020*). However, these tests present significant limitations when applied to PLHIV. TST is often ineffective in detecting *Mycobacterium tuberculosis* infection in PLHIV, especially in individuals with CD4 lymphocyte counts below 200 cells/mm$^3$ (*Fisk et al., 2003*; *Markowitz et al., 1993*). The accuracy of IGRAs in PLHIV is under debate. A systematic review and meta-analysis indicated that IGRAs performed similarly to TSTs in detecting *Mycobacterium tuberculosis* among PLHIV, suggesting potential insensitivity of IGRAs (*Cattamanchi et al., 2011*). On June 8, 2017, the Food and Drug Administration (FDA) granted approval to the QuantiFERON-TB Gold Plus (QFT-Plus), a fourth-generation QFT test, replacing the QuantiFERON-TB Gold In-Tube (QFT-GIT). QFT-Plus represents a new-generation iteration of QFT-GIT and has gained widespread utilization in the diagnosis of *Mycobacterium tuberculosis* infection. The primary distinction between QFT-Plus and QFT-GIT lies in the inclusion of two *Mycobacterium tuberculosis*-specific antigen-coated tubes within QFT-Plus, designated as TB1 and TB2. While TB1 comprises extended peptides from ESAT-6 and CFP-10 (omitting TB7.7), TB2 incorporates six truncated peptides, alongside the same constituents present in TB1. This design prompts both CD4 and CD8 T-cell immune responses (*Nikolova et al., 2013*). CD8+ cytotoxic T cells have emerged as a crucial element in the host's immune response to and regulation of *Mycobacterium tuberculosis*. Research has demonstrated notably heightened CD8+ T-cell reactions in individuals with smear-positive or active pulmonary TB, in comparison to those with smear-negative or *Mycobacterium tuberculosis* infection (*Day et al., 2011*; *Rozot et al., 2013*). Robust CD8+ T-cell responses have also been observed in individuals recently exposed to patients with active tuberculosis infections (*Nikolova et al., 2013*). Additionally, patients coinfected with *Mycobacterium tuberculosis* and PLHIV have exhibited the ability to sustain CD8-positive T-cell antigen responses to *Mycobacterium tuberculosis*, even in the context of diminished CD4+ T-cell counts. In individuals coinfected with *Mycobacterium tuberculosis* and HIV, it has been demonstrated that they can sustain CD8+ T-cell antigen responses to *Mycobacterium tuberculosis* despite having low CD4+ T-cell counts, as evidenced by previous studies (*Sutherland et al., 2010*). QFT-Plus was expected to amplify sensitivity, specifically among individuals with compromised immune

systems. To the best of our knowledge, no direct comparative study has been conducted to evaluate the performance of QFT-GIT and QFT-Plus in diagnosing *Mycobacterium tuberculosis* infection among PLHIV in China. Thus, we designed this cross-sectional study to directly compare the diagnostic performance of QFT-GIT and QFT-Plus for detecting *Mycobacterium tuberculosis* infection in this population. Additionally, we assessed the quantitative changes in IFN-γ levels measured by QFT-GIT after preventive treatment to explore its potential role in monitoring treatment response.

## MATERIALS AND METHODS

### Study population and design

The study was conducted in a central prison hospital in Jiangsu Province, China. This facility is designated for individuals incarcerated within the province who require specialized monitoring and treatment for infectious diseases, including HIV and tuberculosis. As previously described (*Lu et al., 2023*), the hospital provides enhanced healthcare services for inmates diagnosed with HIV, tuberculosis, gonorrhea, syphilis, and similar conditions. This research was embedded within the routine health assessments carried out for provincial prisons, which include blood tests, biochemical analyses, hepatitis serology, syphilis and hepatitis C antibody tests, electrocardiograms, chest X-rays, CD4 cell counts, and HIV viral load measurements. All participants underwent testing with both QFT-GIT and QFT-Plus, and individuals diagnosed with active tuberculosis were excluded to focus on *Mycobacterium tuberculosis* infection. PLHIV identified with *Mycobacterium tuberculosis* infection or at risk of infection based on IGRA results were offered preventive treatments, including chemotherapy and immunotherapy. Participants had the freedom to choose their preferred preventive treatment option. Chemotherapy consisted of a 3-month course of isoniazid and rifapentine (3HP), in accordance with WHO recommendations for managing *Mycobacterium tuberculosis* infection. Immunotherapy was a pilot regimen involving six doses of *Mycobacterium* vaccae administered biweekly, following established guidelines for the management of *Mycobacterium tuberculosis* infection. Participants were classified based on clinical evaluations, IGRA results, and the absence of active tuberculosis symptoms. This approach ensured the study population was representative of PLHIV requiring *Mycobacterium tuberculosis* infection management.

### QFT-GIT and QFT-Plus assay

The QFT-GIT and QFT-Plus assays (Qiagen, Hilden, Germany) were carried out following the manufacturer's guidelines. Within four hours of collecting whole-blood samples in lithium heparin tubes, one milliliter of whole blood was transferred into a *Mycobacterium tuberculosis* antigen tube (TB) for the QFT-GIT test, and into *Mycobacterium tuberculosis* antigen tubes (TB1 and TB2) for the QFT-Plus test, along with separate tubes for nil and mitogen controls. These five tubes were promptly placed in a 37 °C incubator for 16–24 h. Quantitative measurement of IFN-γ was concurrently performed for both assays using a DS-2 automated ELISA processor. Results were considered positive when the IFN-γ concentration in the *Mycobacterium tuberculosis* antigen tube (TB for QFT-GIT and either TB1 or TB2 for QFT-Plus) exceeded the IFN-γ concentration in the nil tube by at least

0.35 IU/ml and was at least 25% of the value in the nil tube. Results were considered indeterminate if the IFN-γ concentration in the nil tube was greater than 8.0 IU/ml, or if the IFN-γ concentration in the mitogen tube was less than 0.5 IU/ml. QFT-GIT test reversion and conversion are defined as changes in the test results from positive to negative and negative to positive, respectively, with a threshold of 0.35 IU/ml.

## Statistical analyses

We employed 2 × 2 contingency tables alongside means accompanied by standard deviations (SD) for the consolidation of continuous and categorical variables. The Fisher exact test or chi-square test was selected for the comparison of the two tests, depending on appropriateness. Quantitative data was compared by Paired-Samples T test. We also conducted a comparison of binary events in QFT-GIT and QFT-Plus using Cohen's kappa (k) coefficient. Kappa coefficients were categorized as follows: poor ($k \leq 0.20$), fair ($0.20 < k \leq 0.40$), moderate ($0.40 < k \leq 0.60$), good ($0.60 < k \leq 0.80$), and very good ($0.80 < k \leq 1.00$). Bivariate correlation was used to compare quantitative IFN-γ levels between QFT-GIT and QFT-Plus.

## Ethics approval and consent to participate

This study underwent thorough review and received approval from the ethics committee at the Jiangsu Provincial Center for Disease Control and Prevention of Jiangsu Province (Ethical ID: JSJK2023-B029-02). All eligible participants provided their informed consent through written documentation.

# RESULTS

## Subject characteristics

A total of 232 PLHIV patients were included in this study, of which 214 (92.2%) had QFT-GIT, EC skin test and TST before. Among 232 PLHIV patients, most of them (87.5%) were males. Median age (interquartile range, IQR) was 48 (40–54) years. About a quarter of individuals had a CD4 count of $\geq$ 500 cells/mm$^3$ with the median CD4 (IQR) 369 (265–500) cells/mm$^3$. The positive rates of initial QFT-GIT, EC skin test and TST test having before were 25.0%, 14.5% and 15.4%, respectively. A total of 47 individuals (74.6%) underwent the 3HP therapy, while 16 individuals (25.4%) received the *Mycobacterium* vaccae treatment (Table 1).

## Detection of *Mycobacterium tuberculosis* infection using QFT-GIT and QFT-Plus

In total, 57 patients (24.6%) were identified with *Mycobacterium tuberculosis* infection based on the outcomes of the QFT-GIT test, while 56 patients (24.1%) were diagnosed for *Mycobacterium tuberculosis* infection using the QFT-Plus. Indeterminate results were observed in one patient (0.4%) by QFT-GIT and 2 (0.9%) by QFT-Plus. Out of the 56 positive results, a substantial 92.9% (52 cases) and an even more impressive 98.2% (55 cases) were found to be positive in TB1 and TB2, respectively. Out of the 57 patients with positive outcomes from the QFT-GIT test, 51 individuals (89.5%) exhibited positive results

Lu et al. (2025), *PeerJ*, DOI 10.7717/peerj.19195

**Table 1  Characteristics of enrolled subjects according to results obtained from QFT-GIT and QFT-Plus ($n = 232$).**

| Characteristic | All | QFT-GIT | | | QFT-Plus | | | | |
|---|---|---|---|---|---|---|---|---|---|
| | | No. (%) | | | No. (%) | | | No. (%) of positive results in | |
| | | Negative | Positive | *P* value | Negative | Positive | *P* value | TB1 | TB2 |
| Median Age, yrs (IQR) | 40 (35–48) | 39 (34–47) | 43 (37–49) | | 40 (34–47) | 42 (37–49) | | 42 (36–49) | 42 (37–49) |
| Sex | | | | | | | | | |
| Female | 29 (12.5) | 17 (60.7) | 11 (39.3) | 0.056 | 18 (64.3) | 10 (35.7) | 0.135 | 9 (17.3) | 10 (17.9) |
| Male | 203 (87.5) | 157 (77.3) | 46 (22.7) | | 156 (77.2) | 46 (22.8) | | 43 (82.7) | 46 (82.1) |
| Median CD4 count, Cell/mm3, (IQR) | 369 (265–500) | 371 (254–502) | 371 (289–497) | | 376 (257–509) | 371 (286–497) | | 373 (298–479) | 369 (288–479) |
| CD4 count | | | | | | | | | |
| <500 | 172 (74.1) | 129 (75.4) | 42 (24.6) | 0.946 | 128 (75.3) | 42 (24.7) | 0.831 | 39 (75.0) | 42 (75.0) |
| ≥500 | 60 (25.9) | 45 (75.0) | 15 (25.0) | | 46 (76.7) | 14 (23.3) | | 13 (25.0) | 14 (25.0) |
| Levels of IFN-γ, IU/ml (IQR) | 0.01 (−0.01, 0.34) | 0.0 (−0.01, 0.01) | 2.51 (1.08, 4.53) | | | | | 0.01 (0.00, 0.02) | 1.71 (0.73, 4.19) |
| Initial QFT-GIT | | | | | | | | | |
| Negative | 156 (75.0) | 148 (94.9) | 8 (5.1) | <0.0001 | 148 (95.5) | 7 (4.5) | <0.0001 | 6 (15.0) | 7 (15.9) |
| Positive | 52 (22.4) | 15 (28.8) | 37 (71.2) | | 15 (28.8) | 37 (71.2) | | 34 (85.0) | 37 (84.1) |
| Initial EC skin test | | | | | | | | | |
| Negative | 183 (85.5) | 166 (90.7) | 17 (9.3) | <0.0001 | 166 (91.2) | 16 (8.8) | <0.0001 | 14 (32.6) | 16 (34.0) |
| Positive | 31 (14.5) | 0 (0.0) | 31 (100.0) | | 0 (0.0) | 31 (100.0) | | 29 (67.4) | 31 (66.6) |
| Initial TST | | | | | | | | | |
| Negative | 181 (84.6) | 158 (88.8) | 20 (11.2) | <0.0001 | 158 (89.3) | 19 (10.7) | <0.0001 | 18 (41.9) | 19 (40.4) |
| Positive | 33 (15.4) | 8 (22.2) | 28 (77.8) | | 8 (22.2) | 28 (77.8) | | 25 (58.1) | 28 (59.6) |

**Notes.**

QFT-GIT, QuantiFERON-TB Gold In-Tube; QFT-Plus, QuantiFERON-TB Gold Plus.

for both TB1 and TB2. However, one patient displayed positive outcomes exclusively for TB1(TB1$^+$TB2$^-$, 1.8%), and three patients for TB2 (TB1$^-$TB2$^+$ 5.3%).

## Concordance between QFT-GIT and QFT-Plus

The QFT-Plus test exhibited concordance with the QFT-GIT in 96.5% (55/57) of positive tests and 98.9% (172/174) of negative tests. Across all subjects, the overall agreement was 98.3% (228/232), with a Cohen's kappa value of 0.954 (95% CI [0.904–0.990]). The consistency rates between QFT-GIT and QFT-Plus TB1 and TB2 were remarkably high, with QFT-GIT achieving a rate of 97.4% (95% CI [94.5%–99.0%]) and QFT-Plus TB1 and TB2 demonstrating an even higher rate of 97.8% (95% CI [95.0%–99.3%]) (Table 2). Indeterminate results were observed in 1 patient in QFT-GIT and 2 in QFT-Plus. As illustrated in Table 3, there were four individuals with HIV displaying varying test outcomes. Among these subjects, two tested positive using QFT-GIT but yielded negative results with QFT-Plus. Conversely, one individual tested positive with QFT-Plus, while another exhibited an indeterminate result, yet tested negative with QFT-GIT. In the spectrum of IFN-γ reactions encompassing four distinct outcomes, the QFT-GIT measurements ranged from 0.06 to 2.44 IU/ml. In contrast, the measurements in QFT-Plus TB1 ranged between −0.03 and 0.13 IU/ml, and in QFT-Plus TB2, the range was 0 to 0.50 IU/ml.

## Assessment of QFT-GIT and QFT-Plus IFN-γ levels

Within 232 individuals, the disparities in absolute quantitative IFN-γ levels between the QFT-GIT TB antigen tube and the QFT-Plus TB1 and TB2 antigen tubes spanned a spectrum of −7.26 IU/ml to 9.83 IU/ml, −0.26 IU/ml to > 10 IU/ml and −0.23 IU/ml to > 10 IU/ml, respectively. The quantitative IFN-γ level observed in QFT-GIT surpassed that of QFT-Plus TB1 ($P = 0.04$), while the difference compared to QFT-Plus TB2 exhibited a marginal trend ($P = 0.134$). The absolute quantitative IFN-γ levels between the QFT-Plus TB1 and TB2 antigen tubes displayed a variance spanning from −5.72 IU/ml to 1.67 IU/ml, with a median discrepancy of 0 IU/ml. In general, a strong correlation was evident among the IFN-γ levels in the QFT-GIT TB antigen tube and both QFT-Plus TB1 and TB2 tubes (Pearson's correlation coefficients 0.867 and 0.839), as illustrated in Fig. 1. The correlation between IFN-γ values was also assessed solely within samples yielding a positive outcome of the QFT-GIT. Despite a slight reduction, the association between QFT-GIT and QFT-Plus IFN-γ levels remained notably robust, indicated by Pearson's coefficients of 0.800 for QFT-GIT *versus* QFT-Plus TB1, 0.722 for QFT-GIT *versus* QFT-Plus TB2, and 0.863 for QFT-Plus TB1 *versus* TB2 tubes.

## Quantitative data of IFN-γ production after preventive treatment

Out of a total of 232 individuals, 63 individuals underwent preventive treatments, with 47 of them receiving the 3HP treatment, while the remaining 16 individuals were administered *Mycobacterium* Vaccae. Among the subgroup of 63 individuals who received preventive treatment, 10 (15.9%) exhibited negative QFT-GIT results, and 11 (17.5%) did not undergo QFT-GIT testing. In the group of 52 individuals who underwent dual QFT-GIT tests, a significant proportion of 23.1% (12 individuals) experienced a revision in their QFT-GIT

**Table 2  Agreement of diagnostic results for QuantiFERON-TB Gold In-Tube and QuantiFERON-TB Gold Plus.**

| Test | QFT-GIT result | | | | |
|---|---|---|---|---|---|
| | Negative | Positive | Indeterminate | Consistency (95% CI) | Kappa value (95% CI) |
| QFT-Plus result: | | | | | |
| Negative | 172 | 1 | 1 | | |
| Positive | 2 | 55 | 0 | 98.3 (95.6, 99.5) | 0.954 (0.904, 0.990) |
| Indeterminate | 0 | 0 | 1 | | |
| QFT-Plus TB1 | | | | | |
| Negative | 173 | 0 | 1 | | |
| Positive | 5 | 52 | 0 | 97.4 (94.5, 99.0) | 0.930 (0.869, 0.977) |
| Indeterminate | 0 | 0 | 1 | | |
| QFT-Plus TB2 | | | | | |
| Negative | 172 | 1 | 1 | | |
| Positive | 3 | 54 | 0 | 97.8 (95.0, 99.3) | 0.943 (0.884, 0.989) |
| Indeterminate | 0 | 0 | 1 | | |

**Notes.**
QFT-GIT, QuantiFERON-TB Gold In-Tube; QFT-Plus, QuantiFERON-TB Gold Plus.

results, shifting from a positive to a negative outcome. Interestingly, within this subgroup, a notable 5.8% (three individuals) demonstrated a QFT-GIT conversion transition from negative to positive as a result of *Mycobacterium* impact. As depicted in Fig. 2, within the cohort of 42 individuals assessed prior to and following treatments, the baseline IFN-$\gamma$ level exhibited notably higher values (median: 5.57; IQR: 0.58, 10.0 IU/ml) in comparison to the levels observed after treatment (median: 1.27; IQR: 0.11, 2.92) ($P < 0.001$). A similar pattern was observed in individuals who initially tested positive for QFT-GIT, with baseline levels (median: 7.40; IQR: 2.58, 10.0 IU/ml) contrasting with post-treatment levels (median: 1.98; IQR: 0.17, 4.05) ($P < 0.001$). But there were no notable distinctions observed in the QFT-GIT, QFT-Plus TB1 and TB2 ($P = 0.178$ and 0.520, respectively, Fig. 2). Also, no significant differences were detected in the reduction of IFN-$\gamma$ levels between the 3HP and *Mycobacterium vaccae* therapies ($P = 0.638$).

## DISCUSSION

In this research, we assessed the diagnostic potential of both QFT-GIT and QFT-Plus in detecting *Mycobacterium tuberculosis* infection across PLHIV in a prison hospital. There was no notable distinction observed in FN-$\gamma$ levels when comparing the QFT-GIT and QFT-Plus assays and the effectiveness of QFT-GIT and QFT-Plus in identifying *Mycobacterium tuberculosis* infection among PLHIV with relatively higher CD4 count levels was found to be comparable, and the correspondence between these two tests within the PLHIV group demonstrated a strong level of concurrence, exceeding 90%. Furthermore, our investigation revealed that among PLHIV regardless of whether it's the chemotherapy regimen or the immunotherapy regimen, preventive treatment for *Mycobacterium tuberculosis* infection would lead to a reduction in IFN-$\gamma$ levels.

Lu et al. (2025), *PeerJ*, DOI 10.7717/peerj.19195

Peer J

**Table 3 Characteristics of discordant QuantiFERON-TB Gold In-Tube and QuantiFERON-TB Gold Plus test among HIV individuals.**

| Subject | CD4 count | QFT-GIT TB Ag-Nil | QFT-GIT Result | QFT-Plus TB1-Nil | QFT-Plus TB2-Nil | QFT-Plus Result | Age | Sex | Initial QFT-GIT TB Ag-Nil | Initial QFT-GIT result | TST result | EC result | BCG |
|---|---|---|---|---|---|---|---|---|---|---|---|---|---|
| 1 | 149 | 0.09 | Negative | 0 | 0.01 | Indeterminate | 58 | Male | 0.13 | Negative | Negative | Negative | Yes |
| 2 | 236 | 0.06 | Negative | 0.1 | 0.5 | Positive | 38 | Male | 0.21 | Negative | Negative | Negative | No |
| 3 | 254 | 2.44 | Positive | −0.03 | 0 | Negative | 40 | Female | 0.01 | Negative | Negative | Negative | Yes |
| 4 | 658 | 1.76 | Positive | 0.13 | 0.16 | Negative | 51 | Male | 0.66 | Negative | Negative | Negative | No |

**Notes.**

QFT-GIT, QuantiFERON-TB Gold In-Tube; QFT-Plus, QuantiFERON-TB Gold Plus.

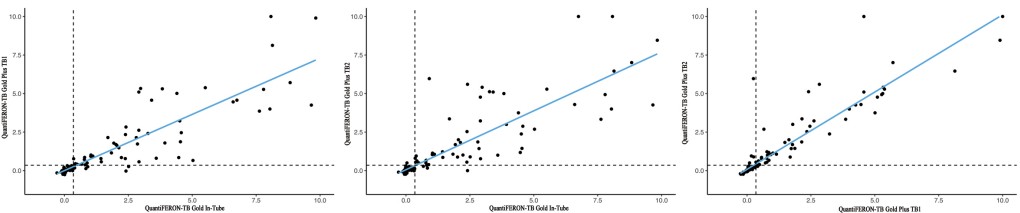

**Figure 1   Linear regression analysis in QuantiFERON-TB Gold In-Tube, QuantiFERON-TB Gold Plus TB1, and TB2 among all subjects.**

A review and meta-evaluation revealed a 1.3% increase in sensitivity for QFT-Plus compared to QFT-GIT. However, additional evaluation is needed to ascertain the sensitivity of QFT-Plus in individuals with compromised immune systems. In prior research, the agreements rates between QFT-GIT and QFT-Plus in immunocompromised individuals were about 94% (*Ryu et al., 2018*; *Xu et al., 2022*). QFT-Plus could potentially offer greater utility in identifying *Mycobacterium tuberculosis* infection among elderly and immunocompromised individuals, particularly in situations involving limited sample sizes (*Chien et al., 2018*; *Kim, Jo & Shim, 2020*). According to a particular study's conjecture, due to the inclusion of two tubes in the QFT-Plus assay, a positive outcome in either tube is categorized as a positive result. This alteration could potentially lead to an augmentation in the proportion of positive findings (*Hogan et al., 2019*). *Xu et al. (2022)* found that among individuals with varying immune statuses, the QFT-Plus assay demonstrated an elevated rate of positive results in comparison to the QFT-GIT assay. Additionally, significant distinctions were observed between the findings of these two assays (*Xu et al., 2022*). Nevertheless, in our study, the prevalence of *Mycobacterium tuberculosis* infection measured by QFT-GIT slightly exceeded that of QFT-Plus, albeit without statistical significance. This trend could potentially be attributed to the fact that all PLHIV included in the study exhibited well-maintained CD4 counts, indicating relatively robust immune systems. QFT-Plus could potentially exhibit a significant positivity rate in blood samples collected from individuals with tuberculosis and co-existing HIV infection, even encompassing those with CD4 T-cell counts as low as 100/μ (*Telisinghe et al., 2017*). Another contributing factor could be the application of preventive treatments for *Mycobacterium tuberculosis* infection among all PLHIV, which might have contributed to the narrowing of the disparity between QFT-GIT and QFT-Plus results.

Several studies have highlighted a substantial increase in IFN-γ levels within TB2 tubes when contrasted with TB and TB1 tubes. Moreover, TB1 tubes have shown noticeably higher IFN-γ levels than TB tubes (*Won et al., 2020*). Notably, some investigations have indicated that there is no statistically significant difference in IFN-γ levels among TB, TB1, and TB2 tubes (*Xu et al., 2022*). Nevertheless, in our study, the IFN-γ levels observed in TB tubes were comparable to those in TB2 tubes, yet they were higher than the levels in TB1 tubes. This discrepancy could potentially be attributed to the omission of the TB7.7 peptide in TB1 tubes as opposed to TB tubes (*Hoffmann et al., 2016*; *Pieterman et al., 2018*).

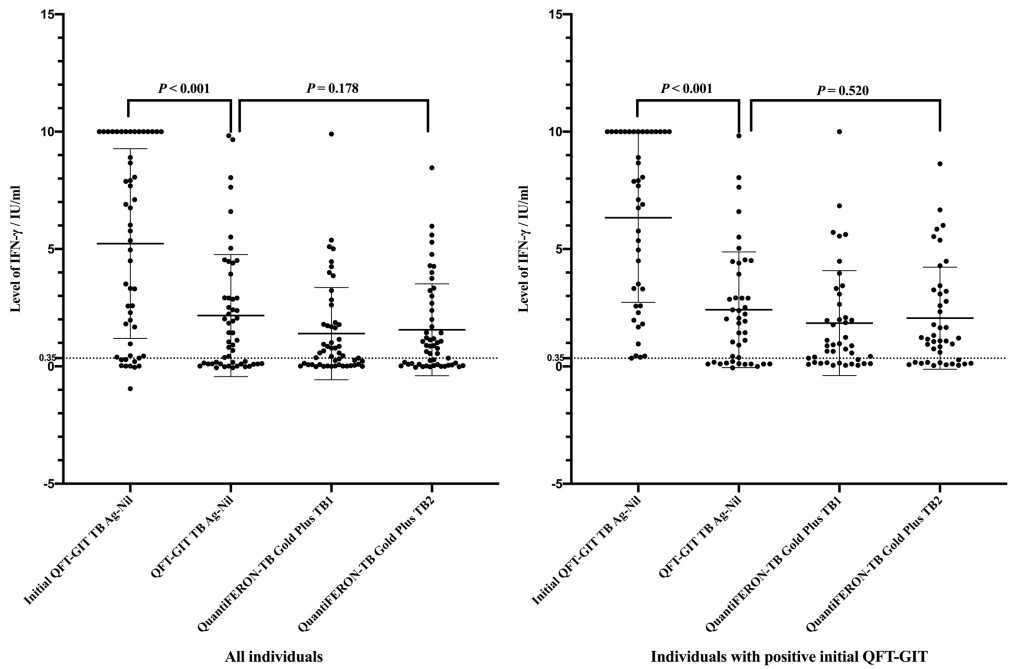

**Figure 2** IFN-γ levels measured using different antigen-containing tubes before and after preventive therapy.

We made another intriguing discovery during our research, revealing a noteworthy trend in the IFN-γ levels of QFT-GIT after the implementation of preventive treatment. Impressively, these levels exhibited a distinct decrease, demonstrating a significant response to the intervention. Additionally, it is noteworthy that approximately 23.1% of the subjects underwent a revision, further underscoring the impact of the preventive measures. Effector T-cell responses following brief incubation with *Mycobacterium tuberculosis*-specific antigens, as employed in custom-made IGRAs, exhibited a correlation with both antigen presence and bacillary burden. This suggests their potential utility as a treatment response marker (*Carrara et al., 2004*). Reversion of IGRA scores following treatment for *Mycobacterium tuberculosis* infection has been noted in various studies. In the most extensive cohort reported to date by *Chee et al. (2007)* among 226 treated close contacts, approximately 38% of the scores reverted based on the manufacturer's criteria for the T-SPOT.TB assay. In a more recent study involving 74 individuals who were QFT-IT-positive tuberculosis contacts, *Lee et al. (2010)* observed a comparable reversion rate of 42%. In our study, the IFN-γ levels of post-treatment was much lower than that of pre-treatment, which could stem from the observation that tuberculosis infection in PLHIV in prison is more prone to being recently acquired like younger subjects (*Goletti et al., 2007*). Furthermore, the decrease in IFN-γ response is achieved with greater ease and rapidity following the treatment of recent contacts who have not been previously exposed to *Mycobacterium tuberculosis* (*Goletti et al., 2007*). However, *Xin et al. (2022)* conducted a comprehensive randomized controlled study involving 910 individuals. These participants

were categorized into three groups: Group A received 8 weeks of once-weekly rifapentine plus isoniazid, Group B received 6 weeks of twice-weekly rifapentine plus isoniazid, and Group C served as untreated controls. Remarkably, their findings indicated comparable rates of persistent QFT-GIT reversion across all three groups ($P = 0.512$). Indeed, in the absence of a parallel control group, the task of determining whether the decline signified a treatment's influence, the organic resolution of the condition, or intrinsic oscillations in T-cell reactions within the same persons became a formidable challenge. These oscillations might arise from elements such as immune senescence or mere variability around the predefined threshold, owing to the reproducibility of the testing procedure (*Banaei, Gaur & Pai, 2016*; *van Zyl-Smit et al., 2009*). Further investigations were required to determine whether reduced IFN-γ levels should be employed as a means to monitor the host's reaction to treatment for *Mycobacterium tuberculosis* infection.

Our study has several limitations that should be acknowledged. Firstly, the absence of a control group limits the robustness of our conclusions, as it restricts our ability to attribute observed effects solely to the interventions studied. Future studies incorporating well-defined control groups are essential to validate our results and strengthen the evidence base. Secondly, we did not conduct the T-SPOT.TB or TST assays concurrently, which could have served as reference standards for assessing the precision of the QFT-GIT and QFT-Plus assays. This omission limits our ability to fully evaluate the comparative accuracy of these assays. Thirdly, we were unable to differentiate whether the decline in IFN-γ levels resulted from preventive treatment, natural clearance, or within-subject variability in T-cell responses, given the absence of a parallel control group. This uncertainty complicates the interpretation of changes in IFN-γ levels over time. Lastly, while IGRA has acknowledged limitations in high-risk settings, we employed it as a practical tool for evaluating treatment efficacy due to the lack of more definitive biomarkers for *Mycobacterium tuberculosis* infection management. This approach, although widely used, introduces inherent uncertainties that should be considered when interpreting our findings.

In conclusion, the diagnostic performance of QFT-GIT and QFT-Plus for detecting *Mycobacterium tuberculosis* infection among PLHIV with relatively higher CD4 counts was comparable. The concordance between these two assays within the PLHIV group exhibited a robust agreement, surpassing the 90% threshold. Furthermore, our investigation revealed that irrespective of whether the treatment involved a chemotherapy regimen or an immunotherapy regimen, preventive *Mycobacterium tuberculosis* infection interventions among PLHIV led to a consistent decrease in IFN-γ levels.

## ACKNOWLEDGEMENTS

The authors thank all investigators from the Jiangsu Provincial Center for Disease Control and Prevention and Jiangsu Prison Administration.

### Funding
This study was funded by the ''Jiangsu Provincial Medical Key Discipline'' (ZDXK202250), ''Jiangsu Provincial Association for Science and Technology Youth Science and Technology Talent Support Project'' (JSTJ-2023-WJ007) and ''Jiangsu Province Preventive Medicine Research Topic Surface Project'' (Ym2023039). The funders had no role in study design, data collection and analysis, decision to publish, or preparation of the manuscript.

### Grant Disclosures
The following grant information was disclosed by the authors:
Jiangsu Provincial Medical Key Discipline: ZDXK202250.
Jiangsu Provincial Association for Science and Technology Youth Science and Technology Talent Support Project: JSTJ-2023-WJ007.
Jiangsu Province Preventive Medicine Research Topic Surface Project: Ym2023039.

### Competing Interests
The authors declare there are no competing interests.

### Author Contributions

- Peng Lu conceived and designed the experiments, performed the experiments, analyzed the data, prepared figures and/or tables, authored or reviewed drafts of the article, and approved the final draft.
- Haitao Yang conceived and designed the experiments, prepared figures and/or tables, authored or reviewed drafts of the article, and approved the final draft.
- Fang Ge conceived and designed the experiments, authored or reviewed drafts of the article, investigate, and approved the final draft.
- Kai Wu conceived and designed the experiments, authored or reviewed drafts of the article, investigate, and approved the final draft.
- Yilin Lian conceived and designed the experiments, authored or reviewed drafts of the article, investigate, and approved the final draft.
- Xiaoyan Ding conceived and designed the experiments, authored or reviewed drafts of the article, investigate, and approved the final draft.
- Jingjing Pan conceived and designed the experiments, authored or reviewed drafts of the article, investigate, and approved the final draft.
- Hui Ding conceived and designed the experiments, authored or reviewed drafts of the article, investigate, and approved the final draft.
- Wei Lu conceived and designed the experiments, authored or reviewed drafts of the article, investigate, and approved the final draft.
- Qiao Liu conceived and designed the experiments, performed the experiments, authored or reviewed drafts of the article, and approved the final draft.
- Limei Zhu conceived and designed the experiments, authored or reviewed drafts of the article, and approved the final draft.

## Human Ethics

The following information was supplied relating to ethical approvals (i.e., approving body and any reference numbers):

The ethics committee at the Jiangsu Provincial Center for Disease Control and Prevention of Jiangsu Province.

## Data Availability

The raw data is available in the Supplemental File.

## Supplemental Information

Supplemental information for this article can be found online at http://dx.doi.org/10.7717/peerj.19195#supplemental-information.

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
