# Peer review of "Performance comparison of QuantiFERON-TB Gold In-Tube and QuantiFERON-TB Gold Plus in detecting Mycobacterium tuberculosis infection among HIV patients in China"

_PeerJ, doi:10.7717/peerj.19195_

## Round 0.1 · original submission · Major Revisions

As you see, the reviewers have raised some serious concerns. You must address these comments properly before the manuscript could be further considered for publication.

Reviewer 1 ·

Basic reporting

Line 115: Mycobacterium Vaccae >> Mycobacterium vaccae
Line 124: Mycobacterium tuberculosis >> Mycobacterium tuberculosis italicize species names. Do these throughout the entire manuscript

Line 218: A review and meta-evaluation revealed a 1.3% increase in sensitivity for QFT-Plus compared to QFT-GIT. Authors need to specifically mention whether the review and meta-evaluation were done in China? Because they indicated that this is the first of such studies in China.

Line 278: In conclusion, the diagnosis performance of QFT-GIT and QFT-Plus across PLHIV with relatively higher CD4 count for Mycobacterium tuberculosis infection was comparable >> In conclusion, the diagnostic performance of QFT-GIT and QFT-Plus for detecting Mycobacterium tuberculosis infection among PLHIV with relatively higher CD4 counts was comparable.

Experimental design

The original primary research's aims follow within the journal's scope.
The research question is well-defined, relevant & meaningful. The authors stated how their research fills an identified knowledge gap. The authors demonstrated that they have performed rigorous investigations to a high technical and ethical standard.

The authors described methods with sufficient detail and information to replicate.

Validity of the findings

The authors demonstrate impact and novelty in this study.

Conclusions are well-stated and linked to the original research question.

Reviewer 2 ·

Basic reporting

The manuscript entitled "Comparative assessment of QuantiFERON-TB Gold In-
Tube and QuantiFERON-TB Gold Plus for diagnosing Mycobacterium Tuberculosis Infection among
persons living with HIV in China" requires extensive rewriting since the language used is unable to convey properly the purpose of this study. The title is misleading since a comparative assessment of new techniques can only be made in comparison to the results used as gold standard in the country of study, with inclusion of proper controls. In the current form, it is incomprehensible to understand the study design and methodology rigorous used by the authors.

Experimental design

The research question is relevant but the study design has major limitations. The recruitment procedure should have been properly defined. If the purpose was comparative assessment of QuantiFERON-TB Gold In-Tube and QuantiFERON-TB Gold Plus for diagnosing Mycobacterium Tuberculosis Infection, why were active TB cases excluded from the study. How were the cases classified? The authors have themselves highlighted the limitations of IGRA and TST for diagnostic purposes in high-risk settings but have then used IGRA as a suggestive marker to detect efficacy of preventative therapy. No explanation has been given why 47 patients were given 3HP treatment and why 16 were given immunotherapy? In absence of a control group, how have the authors interpreted results since all the limitations mentioned are major drawbacks of the study.

Validity of the findings

In absence of clarity of the study design, the validity of the findings cannot be judged. The authors need to explain their work properly.

---

## Round 0.2 · accepted · Accept

Dear Dr. Lu,
I am pleased to inform you that your manuscript has been accepted for publication.

Reviewer 1 ·

Basic reporting

The authors of the manuscript titled "Performance Comparison of QuantiFERON-TB Gold In-Tube and QuantiFERON-TB Gold Plus in Detecting Mycobacterium tuberculosis Infection among HIV Patients in China" have addressed all the comments raised to my satisfaction.

Experimental design

The authors of the manuscript titled "Performance Comparison of QuantiFERON-TB Gold In-Tube and QuantiFERON-TB Gold Plus in Detecting Mycobacterium tuberculosis Infection among HIV Patients in China" have addressed all the comments raised to my satisfaction.

Validity of the findings

None